# *Eed* controls craniofacial osteoblast differentiation and mesenchymal proliferation from the neural crest

Tim Casey-Clyde[1,2,3], S John Liu[1,2,3], Angelo Pelonero[4,5],
Juan Antonio Camara Serrano[6], Camilla Teng[7], Yoon-Gu Jang[7],
Harish N Vasudevan[1,2], Arun Padmanabhan[8,9], Jeffrey Ohmann Bush[7],
David R Raleigh[1,2,3]*

[1]Department of Radiation Oncology, University of California San Francisco, San Francisco, United States; [2]Department of Neurosurgery, University of California San Francisco, San Francisco, United States; [3]Department of Pathology, University of California San Francisco, San Francisco, United States; [4]Gladstone Institutes, San Francisco, United States; [5]Roddenberry Center for Stem Cell Biology at Gladstone Institutes, San Francisco, United States; [6]Helen Diller Family Comprehensive Cancer Center, University of California San Francisco, San Francisco, United States; [7]Department of Cell and Tissue Biology, University of California San Francisco, San Francisco, United States; [8]Department of Medicine, Division of Cardiology, University of California San Francisco, San Francisco, United States; [9]Cardiovascular Resarch Institute, University of California San Francisco, San Francisco, United States

*For correspondence:
david.raleigh@ucsf.edu

Competing interest: The authors declare that no competing interests exist.

## eLife Assessment

In this **valuable** study, the authors used an elegant genetic approach to delete EED at the post-neural crest induction stage. The usage of the single-cell RNA-seq analysis method is extremely suitable to determine changes in the cell type-specific gene expression during development. Results backed by **solid** evidence demonstrate that Eed is required for craniofacial osteoblast differentiation and mesenchymal proliferation after the induction of the neural crest.

**Abstract**    The histone methyltransferase Polycomb repressive complex 2 (PRC2) is required for specification of the neural crest, and mis-regulation of neural crest development can cause severe congenital malformations. PRC2 is necessary for neural crest induction, but the embryonic, cellular, and molecular consequences of PRC2 activity after neural crest induction are incompletely understood. Here, we show that *Eed*, which encodes a protein that is a core subunit of PRC2, is required for craniofacial osteoblast differentiation and mesenchymal proliferation after induction of the neural crest. Integrating mouse genetics with single-cell RNA sequencing and epigenetic profiling, our results reveal that conditional knockout of *Eed* after neural crest cell induction causes severe craniofacial hypoplasia, impaired craniofacial osteogenesis, and attenuated craniofacial mesenchymal cell proliferation that is first evident in post-migratory neural crest cell populations. We show that *Eed* drives mesenchymal differentiation and proliferation in vivo and in primary craniofacial cell cultures by epigenetically regulating diverse transcription factor programs that are required for specification of post-migratory neural crest cells. These data enhance understanding of epigenetic mechanisms that underlie craniofacial development and shed light on the embryonic, cellular, and molecular drivers of rare congenital syndromes in humans.

## Introduction

The embryonic neural crest is a multipotent progenitor cell population that gives rise to peripheral neurons, glia, Schwann cells, melanocytes, and diverse mesenchymal cells such as osteoblasts, chondrocytes, fibroblasts, cardiac mesenchyme, and cardiomyocytes (*Bronner and LeDouarin, 2012*; *Simões-Costa and Bronner, 2015*). Following induction in the neural tube, cranial neural crest cells undergo dorsolateral migration to the pharyngeal arches and differentiate to form craniofacial bones and cartilage (*Minoux and Rijli, 2010*). To do so, cranial neural crest cells express transcription factors such as Sox9, Sox10, and Twist1 that specify cell fate decisions in derivatives of the neural crest (*Bertol et al., 2022*; *Cheung et al., 2005*). Pre-migratory neural crest cell specification is partially controlled by the epigenetic regulator Polycomb repressive complex 2 (PRC2), an H3K27 histone methyltransferase that is broadly responsible for chromatin compaction and transcriptional silencing (*Margueron and Reinberg, 2011*). The catalytic activity of PRC2 is comprised of four core subunits: (1) enhancer of zeste homologue 1 (Ezh1) or Ezh2, (2) suppressor of zeste 12 (Suz12), (3) RBBP4 or RBBP7, and (4) embryonic ectoderm development (Eed) (*Piunti and Shilatifard, 2021*). Eed binds to H3K27 trimethylation peptides (H3K27me3) and stabilizes Ezh2 for allosteric activation of methyltransferase activity and the on-chromatin spreading of H3K27me3. *Eed* is required for stem cell plasticity, pluripotency, and maintaining cell fate decisions, but all PRC2 core subunits are required for embryonic development, and the loss of any individual subunit is embryonic lethal around gastrulation (*Faust et al., 1995*; *O'Carroll et al., 2001*; *Pasini et al., 2004*). *Eed* null embryos can initiate endoderm and mesoderm induction but have global anterior-posterior patterning defects in the primitive streak (*Faust et al., 1995*; *Schumacher et al., 1996*) and genome-wide defects in H3K27me3 (*Montgomery et al., 2007*).

Ezh2 has been studied in the context of pre-migratory neural crest development. In mice, *Wnt1-Cre Ezh2$^{Fl/Fl}$* embryos fail to develop skull and mandibular structures due to de-repression of Hox transcription factors in cranial neural crest cells (*Ferguson et al., 2018*; *Kim et al., 2018*; *Schwarz et al., 2014*), and loss of *Ezh2* in mesenchymal precursor cells causes skeletal defects and craniosynostosis in *Prrx1-Cre Ezh2$^{Fl/Fl}$* embryos (*Dudakovic et al., 2015*). Ezh2 also regulates neural crest specification in Xenopus (*Tien et al., 2015*), and *Col2-Cre Eed$^{Fl/Fl}$* mice have kyphosis and accelerated hypertrophic differentiation due to de-repression of Wnt and TGF-β signaling in chondrocytes (*Mirzamohammadi et al., 2016*). Genetic and molecular interactions allow *Eed* to repress Hox genes to maintain vertebral body identity during mouse development (*Kim et al., 2006*), but the embryonic, cellular, and molecular consequences of *Eed* activity in craniofacial development remain incompletely understood. To address this gap in our understanding of epigenetic mechanisms that may contribute to craniofacial development, we conditionally deleted *Eed* from the migratory neural crest and its derivatives. Our results suggest that *Eed* is required for craniofacial osteoblast differentiation from post-migratory neural crest, mesenchymal stem cell maintenance, and that *Eed* epigenetically regulates diverse transcription factor programs that are required for mesenchymal cell proliferation, differentiation, and osteogenesis. More broadly, these data show that *Eed* is required at early post-migratory stages in neural crest progenitors of craniofacial structures.

## Results

### Loss of Eed after neural crest induction causes severe craniofacial malformations

To determine if *Eed* is required for the development of neural crest derivatives, homozygous floxed *Eed* alleles (*Eed$^{Fl/Fl}$*) (*Yu et al., 2009*) were used to conditionally delete *Eed* following neural crest cell induction using Cre recombinase under the control of the *Sox10* promoter, which is expressed in the migratory neural crest at embryonic day (E)8.75 and results in complete recombination in post-migratory neural crest cells by E10.5 (*Matsuoka et al., 2005*; *Niethamer et al., 2020*). *Sox10-Cre; Eed$^{Fl/Fl}$* mice were not recovered past postnatal day 0, suggesting that loss of *Eed* following induction of the neural crest is embryonic lethal (*Figure 1a*). However, *Sox10-Cre; Eed$^{Fl/Fl}$* embryos were recovered at expected genotypic frequencies from E9.5 to E17.5 (*Figure 1a*), and there were no differences in the penetrance or severity of *Sox10-Cre; Eed$^{Fl/Fl}$* phenotypes whether Cre was maternally or paternally inherited, as has been reported for other phenotypes arising in cells expressing *Sox10-Cre* (*Crispino et al., 2011*; *Luo et al., 2020*). No overt phenotypes were identified at early post-migratory neural crest cell stages E10.5 or E11.5, but craniofacial malformations were seen in *Sox10-Cre; Eed$^{Fl/Fl}$*

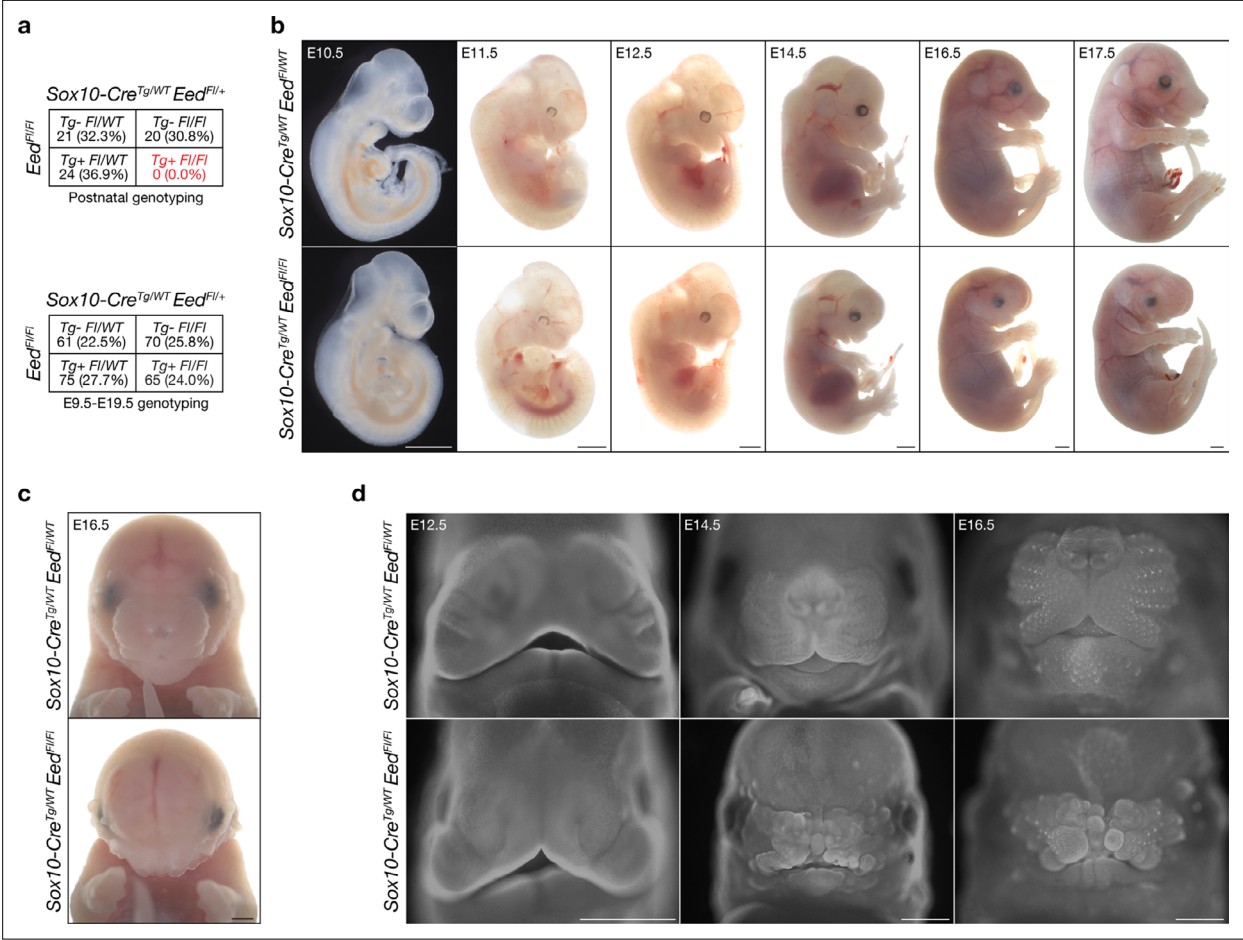

**Figure 1.** *Sox10-Cre; Eed^{Fl/Fl}* embryos develop craniofacial malformations. (**a**) Genotyping frequencies of postnatal (top) and embryonic (bottom) *Sox10-Cre; Eed^{Fl/Fl}* mice. (**b**) Sagittal bright-field images of *Sox10-Cre; Eed^{Fl/WT}* or *Sox10-Cre; Eed^{Fl/Fl}* embryos from E10.5 to E17.5 showing craniofacial malformations appearing at E12.5 that were observed with 100% penetrance. Scale bar, 1 mm. (**c**) Coronal brightfield images of E16.5 *Sox10-Cre; Eed^{Fl/WT}* or *Sox10-Cre; Eed^{Fl/Fl}* embryos showing craniofacial malformations. Scale bar, 1 mm. (**d**) Whole-mount fluorescence microscopy of DAPI-stained *Sox10-Cre; Eed^{Fl/WT}* or *Sox10-Cre; Eed^{Fl/Fl}* heads at E12.5, E14.5, or E16.5. Scale bar, 1 mm. All images are representative of ≥3 biological replicates.

The online version of this article includes the following figure supplement(s) for figure 1:

**Figure supplement 1.** *Sox10-Cre; Eed^{Fl/Fl}* embryos develop craniofacial malformations.

**Figure supplement 2.** *Sox10-Cre; Eed^{Fl/Fl}* embryos develop subtle cardiac malformations.

embryos starting at E12.5 that increased in severity throughout the remainder of embryonic development (*Figure 1b*). *Sox10-Cre; Eed^{Fl/Fl}* phenotypes were broadly consistent with impaired development of craniofacial structures (*Ferguson et al., 2018*; *Schwarz et al., 2014*), including frontonasal and mandibular hypoplasia, a prominent telencephalon and exencephaly resulting from underdevelopment of the viscerocranium and frontal calvarium, collapsed nasal cavity bones, and microtia compared to controls (*Figure 1c*). *Sox10-Cre; Eed^{Fl/Fl}* embryos had increased pupillary distance and craniofacial width (*Figure 1—figure supplement 1a–d*), and whole-mount DAPI imaging of *Sox10-Cre; Eed^{Fl/Fl}* embryos revealed severe frontonasal dysplasia with underdeveloped midface and mandible, and an irregular and corrugated facial structure compared to controls (*Figure 1d*). Although PRC2 is ubiquitously expressed in the developing embryo (*O'Carroll et al., 2001*), *Sox10-Cre; Eed^{Fl/Fl}* embryos did not have cardiac outflow tract defects or other structural heart malformations that can arise from impaired neural crest differentiation (*Lindsay et al., 2001*; *Lindsay et al., 1999*; *Nakamura et al., 2009*; *Figure 1—figure supplement 2a–d*; *Videos 1 and 2*). Instead, fetal echocardiography showed subtle structural changes but preserved ejection fraction and fractional shortening in *Sox10-Cre; Eed^{Fl/Fl}* embryos compared to controls (*Figure 1—figure supplement 2d*).

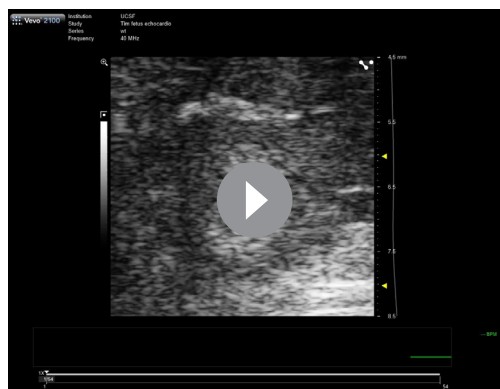

**Video 1.** B-mode echocardiography movie of an embryonic day (E)16.5 *Sox10-Cre; Eed*^Fl/WT heart, transversal view at medium level. Both ventricles, external walls, and interventricular septum are visible. Right ventricle is up, and left ventricle is down. The difference in size between ventricles is visible. The ventricular contraction is rhythmic, symmetric, and complete.

https://elifesciences.org/articles/100159/figures#video1

# Eed regulates craniofacial mesenchymal cell differentiation and proliferation from the early post-migratory neural crest

Skeletal stains, micro-computed tomography (microCT), histology, and immunofluorescence were used to define how loss of *Eed* impacts craniofacial development. Whole-mount Alcian blue/Alizarin Red staining of *Sox10-Cre; Eed*^Fl/Fl embryos revealed hypoplasia of the viscerocranium, including reduced frontal, temporal, maxillary, and mandibular bones, and complete loss of the tympanic ring and premaxillary and nasal bones compared to controls (*Figure 2a,b*, *Figure 2—figure supplement 1a*). MicroCT validated these findings, showing fragmentation and hypoplasia of the frontal, temporal, maxillary, and mandibular bones, loss of frontal calvarial fusion, and absence of tympanic ring and nasal bones (*Figure 2c–e* and *Videos 3 and 4*). H&E histology showed that frontal calvarium reduction resulted in anteriorly displaced brain structures in *Sox10-Cre; Eed*^Fl/Fl embryos compared to controls,

with the midbrain and cortex in the same coronal plane as the nasal cavity and developing mandible (*Figure 3a*, *Figure 2—figure supplement 1b*). Most of the midfacial structures and mandible were absent in *Sox10-Cre; Eed*^Fl/Fl embryos, and the tongue and masseter muscles were underdeveloped compared to controls (*Figure 3a*).

Immunofluorescence demonstrated disorganized ALPL and Osteocalcin, markers of osteoblasts and mineralizing bone (*Liu et al., 2018*), in craniofacial tissues from *Sox10-Cre; Eed*^Fl/Fl embryos compared to controls (*Figure 3b*, *Figure 2—figure supplement 1c*). Immunofluorescence for Sox9, a marker of chondrocytes that are derived from the neural crest (*Mori-Akiyama et al., 2003*; *Yan et al., 2002*), was also disorganized in *Sox10-Cre; Eed*^Fl/Fl embryos compared to controls (*Figure 3b*). There was a small increase in immunofluorescence labeling index for cleaved Caspase 3 (*Figure 2—figure supplement 1d and e*), suggesting that apoptosis may play a subtle role in craniofacial phenotypes from *Sox10-Cre; Eed*^Fl/Fl embryos compared to controls. In contrast to *Wnt1-Cre Ezh2*^Fl/Fl embryos (*Ferguson et al., 2018*; *Schwarz et al., 2014*), the craniofacial region of *Sox10-Cre; Eed*^Fl/Fl embryos had marked decreased immunofluorescence labeling index for Ki67 (*Figure 3c and d*), a marker of cell proliferation (*Gerdes et al., 1984*), and decreased immunofluorescence staining for Vimentin (Vim), a marker of mesenchymal cells (*Mendez et al., 2010*; *Figure 3c*). Immunofluorescence for Runx2, a transcription factor that is required for osteoblast differentiation and proliferation (*Kawane et al., 2018*; *Komori, 2009*), was also decreased in craniofacial tissues from *Sox10-Cre; Eed*^Fl/Fl embryos compared to controls (*Figure 3e*, *Figure 2—figure supplement 1f*). In support of these findings, (1) BrdU labeling index was decreased in the craniofacial region of *Sox10-Cre; Eed*^Fl/Fl embryos compared to controls (*Figure 3—figure supplement 1a and b*), (2) immunofluorescence for Eed, Runx2, Ki67, and

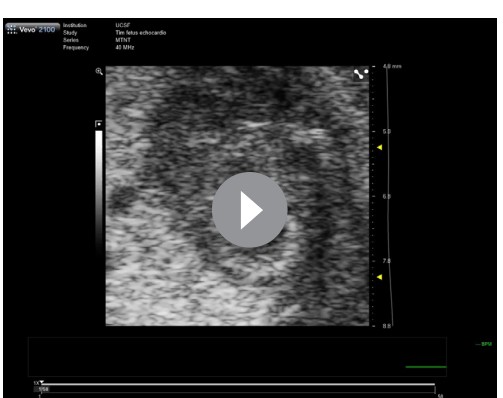

**Video 2.** B-mode echocardiography movie of an embryonic day (E)16.5 *Sox10-Cre; Eed*^Fl/Fl heart, transversal view at medium level. Both ventricles, external walls, and interventricular septum are visible. Right ventricle is up, and left ventricle is down. The ventricular contraction is rhythmic and symmetric.

https://elifesciences.org/articles/100159/figures#video2

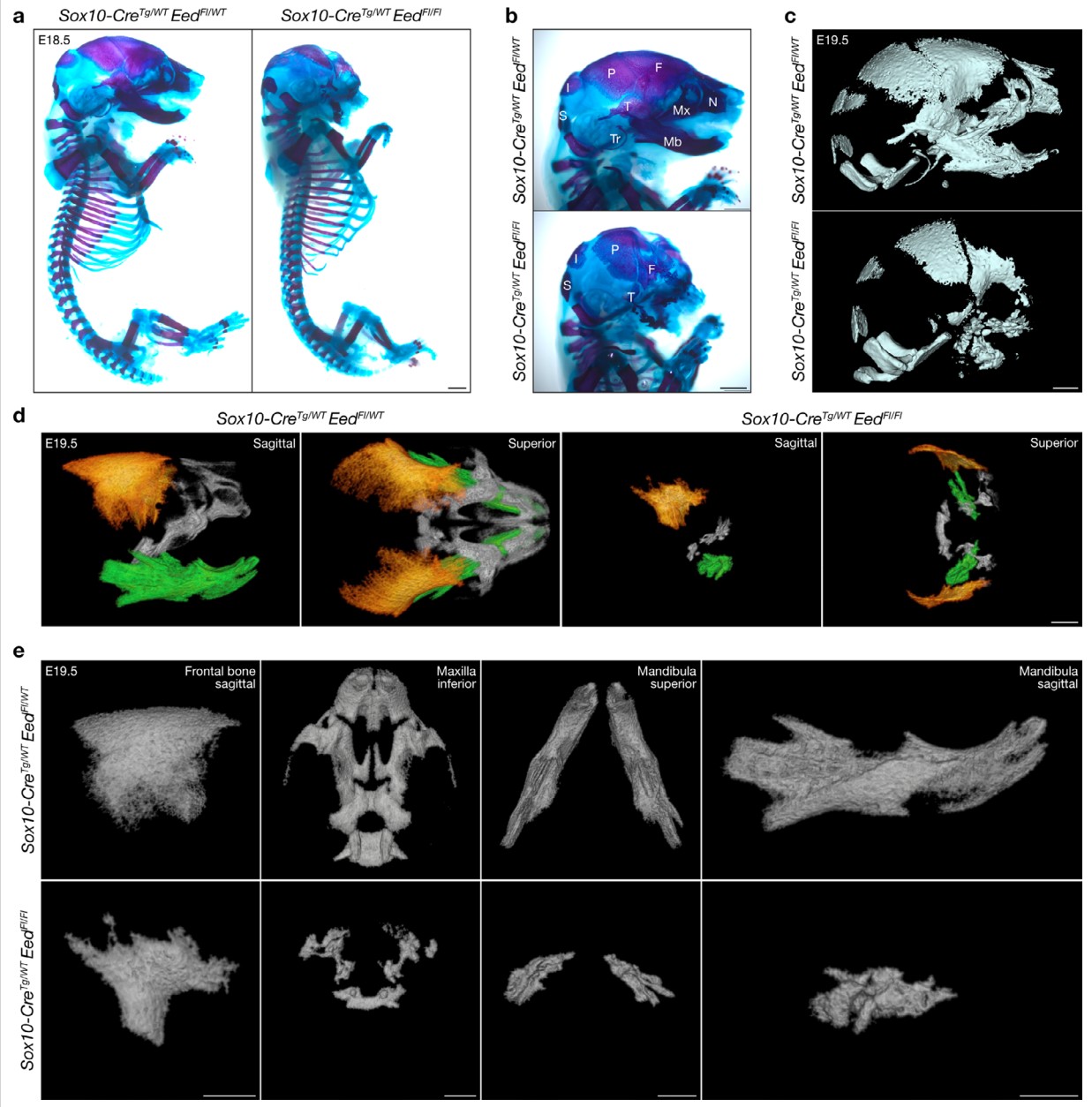

**Figure 2.** Eed is required for craniofacial skeletal development from the neural crest. (**a**) Sagittal bright-field images of whole-mount skeletal stains of embryonic day (E)18.5 *Sox10-Cre; Eed*[Fl/WT] or *Sox10-Cre; Eed*[Fl/Fl] embryos. Alcian blue and Alizarin Red identify cartilage or bone, respectively. Scale bar, 1 mm. (**b**) Magnified sagittal bright-field images of whole-mount skeletal stains of E18.5 *Sox10-Cre; Eed*[Fl/WT] or *Sox10-Cre; Eed*[Fl/Fl] embryos. Bone structures are annotated (I, interparietal; P, parietal; Mx, maxilla; Mb, mandible; Tr, tympanic ring; F, frontal; N, nasal; T, temporal; S, supraoccipital). Scale bar, 1 mm. (**c**) Micro-computed tomography (microCT) images of E19.5 *Sox10-Cre; Eed*[Fl/WT] or *Sox10-Cre; Eed*[Fl/Fl] heads. Scale bar, 1 mm. (**d**) MicroCT images of E19.5 *Sox10-Cre; Eed*[Fl/WT] or *Sox10-Cre; Eed*[Fl/Fl] neural crest-derived craniofacial bones shaded in orange (frontal), gray (maxilla), or green (mandible). Scale bar, 1 mm. (**e**) Magnified microCT images of E19.5 *Sox10-Cre; Eed*[Fl/WT] or *Sox10-Cre; Eed*[Fl/Fl] neural crest-derived craniofacial bones. Scale bars, 1 mm.

The online version of this article includes the following figure supplement(s) for figure 2:

**Figure supplement 1.** *Sox10-Cre; Eed*[Fl/Fl] embryos develop significant craniofacial mesenchymal differentiation phenotypes and subtle apoptotic phenotypes.

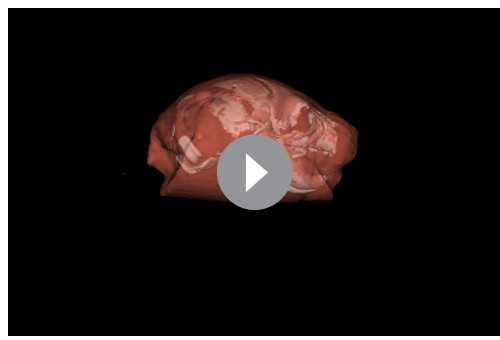

**Video 3.** Three-dimensional reconstruction of micro-computed tomography (microCT) imaging of an embryonic day (E)19.5 *Sox10-Cre; Eed*^Fl/WT head. https://elifesciences.org/articles/100159/figures#video3

ALPL was decreased in primary *Sox10-Cre; Eed*^Fl/Fl craniofacial cell cultures compared to controls (*Figure 3f*, *Figure 3—figure supplement 1c*), and (3) there was no change in immunofluorescence for Sox10 in either primary *Sox10-Cre; Eed*^Fl/Fl craniofacial cell cultures or *Sox10-Cre; Eed*^Fl/Fl embryos compared to controls (*Figure 3—figure supplement 1c and e*). These results suggest that craniofacial mesenchymal differentiation, proliferation, and osteogenesis are impaired in *Sox10-Cre; Eed*^Fl/Fl embryos compared to controls (*Honoré et al., 2003*; *Kim et al., 2003*).

## Eed regulates craniofacial mesenchymal stem cell, osteoblast, and proliferating mesenchymal cell fate from the early post-migratory neural crest

To define cell types and gene expression programs underlying craniofacial phenotypes in *Sox10-Cre; Eed*^Fl/Fl embryos, single-cell RNA sequencing was performed on litter-matched E12.5 *Sox10-Cre; Eed*^Fl/WT and *Sox10-Cre; Eed*^Fl/Fl

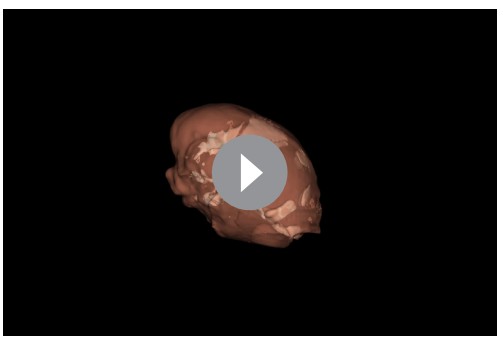

**Video 4.** Three-dimensional reconstruction of micro-computed tomography (microCT) imaging of an embryonic day (E)19.5 *Sox10-Cre; Eed*^Fl/Fl head. https://elifesciences.org/articles/100159/figures#video4

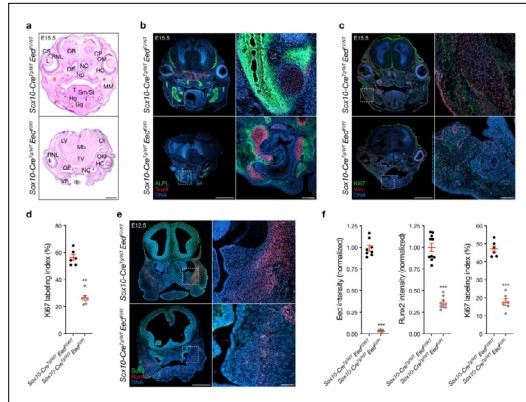

**Figure 3.** Eed regulates craniofacial differentiation and proliferation from the neural crest. (**a**) Bright-field coronal H&E images of E15.5 *Sox10-Cre; Eed*^Fl/WT or *Sox10-Cre; Eed*^Fl/Fl heads. Anatomic structures are annotated (Ct, cortex; CP, cartilage primordium; CS, conjunctival sac; Gg, genioglossus muscle; HC, hyaloid cavity; Hg, hyoglossus muscle; L, lens; LV, lateral ventricle; Mb, midbrain; MM, masseter muscle; Np, nasopharynx; NC, nasal cavity; OB, olfactory bulb; OE, olfactory epithelium; OM, ocular muscle; RNL, retina neural layer; Sm/St, sublingual and submandibular ducts; T, tongue; TV, third ventricle; VF, vibrissa follicles). Scale bar, 1 mm. (**b**) Coronal immunofluorescence images of E15.5 *Sox10-Cre; Eed*^Fl/WT or *Sox10-Cre; Eed*^Fl/Fl heads stained for ALPL (green) and Sox9 (red). Scale bars, 1 mm and 100 µm. (**c**) Coronal immunofluorescence images of E15.5 *Sox10-Cre; Eed*^Fl/WT or *Sox10-Cre; Eed*^Fl/Fl heads stained for Ki67 (green) and Vimentin (red). Scale bars, 1 mm and 100 µm. (**d**) Quantification of Ki67 immunofluorescence labeling index from E15.5 heads. (**e**) Coronal immunofluorescence images of E12.5 *Sox10-Cre; Eed*^Fl/WT or *Sox10-Cre; Eed*^Fl/Fl heads stained for Runx2 (red) and Sox9 (green). Intracranial tissues demonstrate ex vacuo dilation and anterior herniation in *Sox10-Cre; Eed*^Fl/Fl heads. Scale bars, 500 µm and 100 µm. (**f**) Quantification of immunofluorescence imaging intensity from primary craniofacial cell cultures from E12.5 *Sox10-Cre; Eed*^Fl/WT or *Sox10-Cre; Eed*^Fl/Fl embryos. All images are representative of ≥3 biological replicates. DNA is marked by DAPI. Lines represent means and error bars represent standard error of the means. Student's t-tests, **p≤0.01, ***p≤0.0001.

The online version of this article includes the following figure supplement(s) for figure 3:

**Figure supplement 1.** Eed regulates craniofacial differentiation and proliferation in vivo and in primary cell cultures.

heads (n=3 biological replicates per genotype). Uniform manifold approximation and projection (UMAP) analysis of 63,730 single-cell transcriptomes revealed 23 cell clusters (C0-C22) that

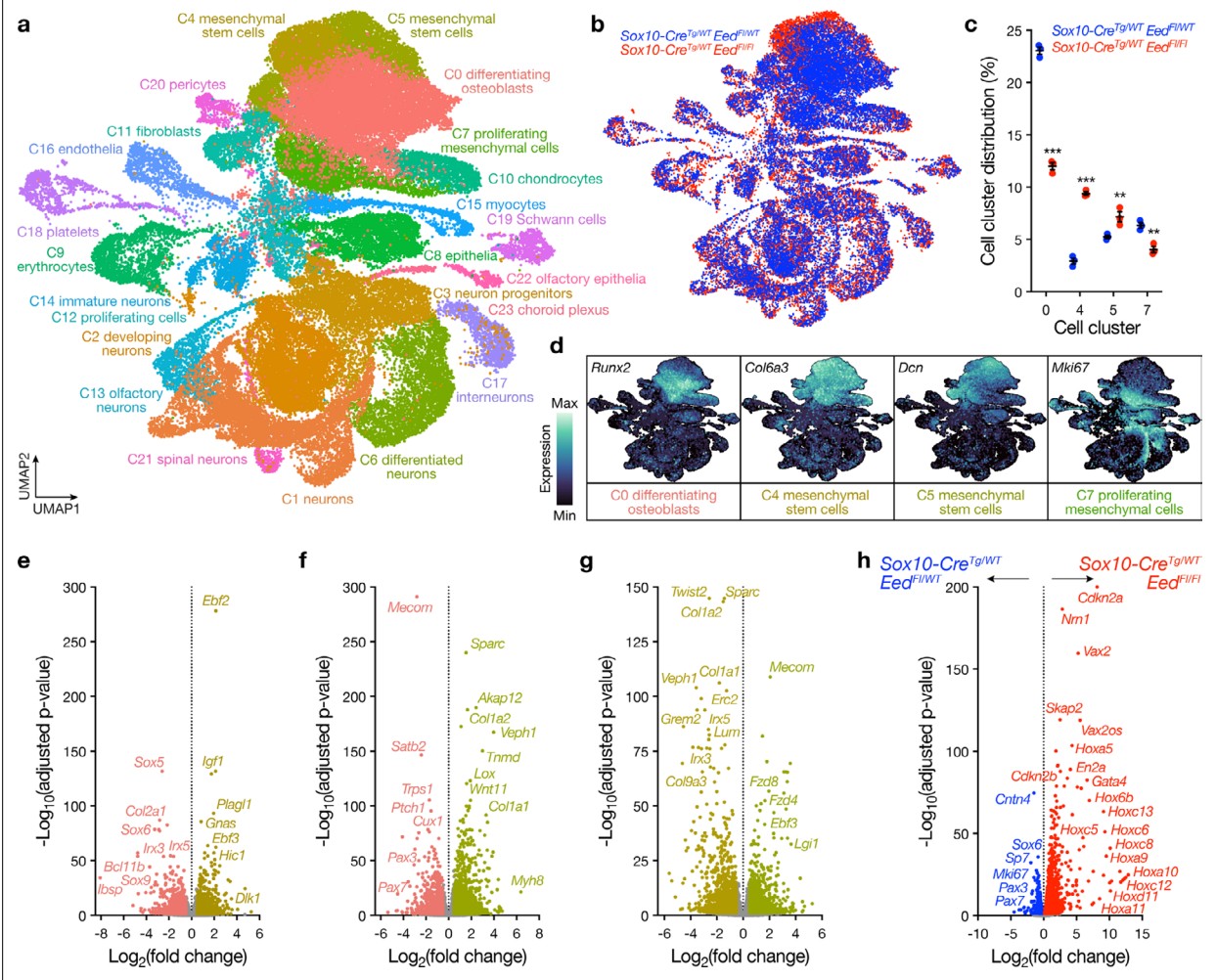

**Figure 4.** Eed regulates craniofacial mesenchymal stem cell, osteoblast, and proliferating mesenchymal cell fate from the neural crest. (**a**) Integrated uniform manifold approximation and projection (UMAP) of 63,730 transcriptomes from single-cell RNA sequencing of litter-matched embryonic day (E)12.5 *Sox10-Cre; Eed*$^{Fl/WT}$ (n=3) or *Sox10-Cre; Eed*$^{Fl/Fl}$ (n=3) heads. (**b**) Integrated UMAP overlaying cell cluster distribution from single-cell RNA sequencing. (**c**) Quantification of mesenchymal cell cluster distribution from single-cell RNA sequencing. Lines represent means and error bars represent standard error of the means. Student's t-tests, **p≤0.01, ***p≤0.0001. (**d**) Gene expression feature plots of mesenchymal lineages from single-cell RNA sequencing. (**e**) Volcano plot showing differentially expressed genes between C0 (osteoblasts) and C4 (mesenchymal stem cells). (**f**) Volcano plot showing differentially expressed genes between C0 and C5 (mesenchymal stem cells). (**g**) Volcano plot showing differentially expressed genes between two clusters of mesenchymal stem cells from single-cell RNA sequencing of litter-matched E12.5 *Sox10-Cre; Eed*$^{Fl/WT}$ (n=3) or *Sox10-Cre; Eed*$^{Fl/Fl}$ (n=3) heads. (**h**) Volcano plot showing differentially expressed genes in differentiating osteoblasts (C0), mesenchymal stem cells (C4, C5), and proliferating mesenchymal cells (C7) from single-cell RNA sequencing of litter-matched E12.5 *Sox10-Cre; Eed*$^{Fl/WT}$ (n=3) versus *Sox10-Cre; Eed*$^{Fl/Fl}$ (n=3) heads.

The online version of this article includes the following figure supplement(s) for figure 4:

**Figure supplement 1.** Cell clusters from single-cell RNA sequencing of embryonic day (E)12.5 *Sox10-Cre; Eed*$^{Fl/WT}$ or *Sox10-Cre; Eed*$^{Fl/Fl}$ heads.

**Figure supplement 2.** Cell cluster marker genes from single-cell RNA sequencing of embryonic day (E)12.5 *Sox10-Cre; Eed*$^{Fl/WT}$ or *Sox10-Cre; Eed*$^{Fl/Fl}$ heads.

**Figure supplement 3.** Mesenchymal cell cluster distribution from single-cell RNA sequencing.

**Figure supplement 4.** Single-cell transcriptome differential expression analyses among mesenchymal cell clusters.

were defined using automated cell-type classification (*Ianevski et al., 2022*), cell signature genes, cell cycle analysis, and differentially expressed cluster marker genes (*Figure 4a*, *Figure 4—figure supplements 1–3*, and *Supplementary file 1 and 2*). Differentiating osteoblasts marked by *Runx2* (C0) and proliferating mesenchymal cells marked by Ki67 (C7) were enriched in *Sox10-Cre; Eed*$^{Fl/WT}$ samples (*Figure 4b–d*, *Figure 4—figure supplement 3*). Mesenchymal stem cells marked by *Col6a3* (C4) or *Dcn* (C5) were enriched in *Sox10-Cre; Eed*$^{Fl/Fl}$ samples (*Figure 4b-d*, *Figure 4—figure supplement*

3), suggesting that loss of *Eed* prevents craniofacial mesenchymal stem cell differentiation (*Jang et al., 2016*; *Lamandé et al., 2006*). There were also subtle differences in the number of interneurons (C17, 1.4% vs 1.9% of cells, p=0.01), Schwann cells (C19, 1.0 vs 1.3% of cells, p=0.03), pericytes (C20, 0.7% vs 0.9%, p=0.05), and spinal neurons (C21, 0.5% vs 0.9% of cells, p=0.003) in *Sox10-Cre; Eed^{Fl/WT}* versus *Sox10-Cre; Eed^{Fl/Fl}* samples, but each of these comprised a minority of recovered cell types (Student's t-tests) (*Supplementary file 1*). There were no differences between genotypes in the number of chondrocytes (C10, 3.1% vs 2.7%, p=0.09), fibroblasts (C11, 2.6% vs 2.7%, p=0.24), endothelia (C16, 1.9% vs 1.6%, p=0.22), hematopoietic cells, or other recovered cell types (Student's t-tests) (*Supplementary file 1*).

Differential expression analysis of single-cell transcriptomes from differentiating osteoblasts (C0), which were enriched in *Sox10-Cre; Eed^{Fl/WT}* samples (*Figure 4c*), compared to mesenchymal stem cells marked by *Col6a3* (C4), which were enriched in *Sox10-Cre; Eed^{Fl/Fl}* samples (*Figure 4c*), showed *Sox5* and *Sox6* were reduced in mesenchymal stem cells (*Figure 4e*, *Figure 4—figure supplement 4*, and *Supplementary file 3*). These data are consistent with the known role of Sox transcription factors in craniofacial specification and development (*Smits et al., 2001*). Iroquois homeobox (Irx) transcription factors 3 and 5, which contribute to craniofacial osteogenesis and mineralization (*Bonnard et al., 2012*; *Cain et al., 2016*; *Tan et al., 2020*), were also suppressed in mesenchymal stem cells compared to differentiating osteoblasts (*Figure 4e*, *Figure 4—figure supplement 4a*, and *Supplementary file 3*).

Differential expression analysis of single-cell transcriptomes from differentiating osteoblasts compared to mesenchymal stem cells marked by *Dcn* (C5), which were also enriched in *Sox10-Cre; Eed^{Fl/Fl}* samples (*Figure 4c*), showed *Mecom*, *Trps1*, and *Ptch1* were suppressed, and *Sparc* and *Tnmd* were enriched in mesenchymal stem cells (*Figure 4f*, *Figure 4—figure supplement 4b*, and *Supplementary file 4*). Loss of *Mecom* causes craniofacial malformations in mice and zebrafish (*Shull et al., 2020*), *Ptch1* is a key component of the Hedgehog pathway that is crucial for craniofacial osteogenesis (*Jeong et al., 2004*; *Metzis et al., 2013*), and *Trps1* regulates secondary palate and vibrissa development (*Cho et al., 2019*; *Fantauzzo and Christiano, 2012*). Consistently, *Sox10-Cre; Eed^{Fl/Fl}* samples had decreased vibrissa compared to controls (*Figure 1d*, *Figure 2—figure supplement 1b*). *Sparc* drives morphogenesis of the pharyngeal arches and inner ear (*Rotllant et al., 2008*), and *Tnmd* regulates mesenchymal differentiation (*Shukunami et al., 2006*). *Pax3* and *Pax7*, key regulators of neural crest migration and development of skeletal structures (*Maczkowiak et al., 2010*), were also suppressed in mesenchymal stem cells compared to differentiating osteoblasts (*Figure 4f*, *Figure 4—figure supplement 4b*, and *Supplementary file 4*).

Differential expression analysis of single-cell transcriptomes from the two clusters of mesenchymal stem cells that were marked by *Col6a3* (C4) versus *Dcn* (C5), both of which were enriched in *Sox10-Cre; Eed^{Fl/Fl}* samples (*Figure 4c*), showed *Twist2*, *Irx3*, and *Irx5*, which regulate mesenchymal differentiation to osteoblasts (*Lee et al., 2000*; *Tamamura et al., 2017*; *Tan et al., 2020*), distinguished mesenchymal stem cell populations (*Figure 4g*, *Figure 4—figure supplement 4c*, and *Supplementary file 5*).

To determine if there were differences in the gene expression programs of differentiating osteoblasts (C0), mesenchymal stem cells (C4, C5), or proliferating mesenchymal cells (C7) in *Sox10-Cre; Eed^{Fl/WT}* versus *Sox10-Cre; Eed^{Fl/Fl}* samples, differential expression analysis was performed on all single-cell transcriptomes from these clusters between genotypes (*Supplementary file 6*). Consistent with craniofacial findings after loss of *Ezh2* (*Ferguson et al., 2018*; *Kim et al., 2018*; *Schwarz et al., 2014*) and vertebral body findings after loss of *Eed* (*Kim et al., 2006*), Hox transcription factors were broadly de-repressed in mesenchymal cell populations from *Sox10-Cre; Eed^{Fl/Fl}* samples compared to controls (*Figure 4h*). PRC2 targets Sp7 in bone marrow stroma cells (*Liu et al., 2013*) and regulates Wnt signaling in dental stem cells (*Jing et al., 2016*), and differential expression analysis also showed that *Sp7* was enriched in mesenchymal single-cell transcriptomes from *Sox10-Cre; Eed^{Fl/WT}* samples and that the Wnt and stem cell regulator gene *Gata4* was enriched in mesenchymal single-cell transcriptomes from *Sox10-Cre; Eed^{Fl/Fl}* samples (*Figure 4h*). *Mki67*, *Pax3*, and *Pax7* were enriched in mesenchymal single-cell transcriptomes from *Sox10-Cre; Eed^{Fl/WT}* samples, and the cell cycle inhibitors *Cdkn2a* and *Cdkn2b* were enriched in mesenchymal single-cell transcriptomes from *Sox10-Cre; Eed^{Fl/Fl}* samples (*Figure 4h*). In sum, these data show that *Eed* specifies craniofacial osteoblast differentiation and mesenchymal cell proliferation by regulating diverse transcription factor programs required for the specification of post-migratory neural crest cells, and that loss of *Eed* inhibits mesenchymal stem

cell differentiation and mesenchymal cell proliferation. These data contrast with the function of *Eed* in chondrocytes, where genetic inactivation of *Eed* at a later stage in mesenchymal development accelerates hypertrophic differentiation, leading to hypoxia and cell death (***Mirzamohammadi et al., 2016***).

## Eed regulates H3K27me3 deposition at transcription factor loci that are required for craniofacial development

To determine if genome-wide methylation changes underlie phenotypes and gene expression changes after *Eed* deletion in post-migratory neural crest cells, H3K27me3 CUT&Tag chromatin profiling was performed on nuclei harvested from E12.5 or E16.5 *Sox10-Cre; Eed*$^{Fl/WT}$ or *Sox10-Cre; Eed*$^{Fl/Fl}$ craniofacial tissues (n=3 biological replicates per genotype per timepoint). At E12.5, conditional *Eed* deletion was associated with global decreases in H3K27me3 deposition and also decreased H3K27me3 deposition at differentially expressed mesenchymal cell cluster genes from single-cell RNA sequencing (C0, C4, C5, C7) (***Figure 5a***, ***Supplementary file 6***). Decreased H3K27me3 deposition was observed with conditional *Eed* deletion at E16.5, albeit to a lesser extent (***Figure 5b***), likely from relative loss of Sox10+ cells that failed to differentiate or proliferate during development without *Eed* and compensatory colonizing of the craniofacial region by Sox10- Eed+ cells.

The gene-specific impact of *Eed*-dependent H3K27me3 deposition was assessed by ranking genes from CUT&Tag chromatin profiling based on relative H3K27me3 signal for identification of methylation-rich regions (MRRs). There were 115 MRRs in *Sox10-Cre; Eed*$^{Fl/WT}$ and 111 MRRs in *Sox10-Cre; Eed*$^{Fl/Fl}$ craniofacial tissues at E12.5 (72 overlapping, 31.9%) and 147 MRRs in *Sox10-Cre; Eed*$^{Fl/WT}$ and 123 MRRs in *Sox10-Cre; Eed*$^{Fl/Fl}$ craniofacial tissues at E16.5 (105 overlapping, 38.9%) (***Figure 5c and d***, ***Supplementary files 7–10***). Ontology analyses of MRRs from *Sox10-Cre; Eed*$^{Fl/WT}$ or *Sox10-Cre; Eed*$^{Fl/Fl}$ craniofacial tissues showed widespread changes in the H3K27me3 deposition at genes involved in transcriptional regulation and organ development and differentiation (***Figure 5—figure supplement 1***).

Transcription factor loci involved in differentiation and development of craniofacial progenitor cell populations were aberrantly methylated in *Sox10-Cre; Eed*$^{Fl/Fl}$ craniofacial tissues compared to controls, including decreased H3K27me3 signal at *Alx1*, *Alx3*, *Barx1*, *Pax7*, and *Tfap2a* loci (***Figure 5c and d***, ***Supplementary file 7–10***). Mis-regulation of *Alx1* and *Alx3* expression results in craniofacial dysplasia (***Pini et al., 2020***), *Barx1* controls skeletal precursors by promoting cartilage fates (***Nichols et al., 2013***), *Pax7* expression is critical for craniofacial progenitor cell fate (***Murdoch et al., 2012***), and *Tfap2a* influences patterning of the craniofacial skeleton through homeobox gene expression (***Van Otterloo et al., 2022***; ***Van Otterloo et al., 2018***). Loci that increased in H3K27me3 signal in *Sox10-Cre; Eed*$^{Fl/Fl}$ craniofacial tissues compared to controls resided near genes such as *Twist1* and *Bmp4*, which are required for stabilizing neural crest cell signatures to guide migrating potential (***Fan et al., 2021***; ***Goodnough et al., 2016***) and are expressed in the frontal primordium at E12.5 by Msx1 and Msx2 to control craniofacial neural crest cell differentiation (***Han et al., 2007***), respectively.

To determine if locus-specific H3K27me3 loss translated to increased gene expression, as predicted from functional deletion of PRC2, single-cell RNA sequencing and CUT&Tag data from E12.5 embryos were integrated. Differentially enriched genes in mesenchymal cell clusters (C0, C4, C5, C7) from single-cell RNA sequencing of *Sox10-Cre; Eed*$^{Fl/Fl}$ samples compared to controls included the developmental transcription factors *Vax2* (Ventral Anterior Homeobox 2) and *Foxa1*, *Foxa2*, and *Foxd3* (***Supplementary file 6***), each of which had gene body or distal element loses in H3K27me3 signal from CUT&Tag chromatin profiling (***Figure 5e***). Forkhead box (FOX) transcription factors are required for craniofacial bone and cartilage development, and FOX mis-regulation expands cartilage domains and inhibits bone fate decisions (***Mundell and Labosky, 2011***; ***Xu et al., 2018***). These data suggest that Vax2, Foxa1, Foxa2, or Foxd3 may contribute to craniofacial phenotypes observed in *Sox10-Cre; Eed*$^{Fl/Fl}$ embryos downstream of H3K27me3 deposition in response to functional deletion of PRC2. More broadly, these data suggest that loss of H3K27me3 signal at developmental gene loci may correlate with gene expression changes that result in aberrant cell fate decisions of craniofacial mesenchymal stem cells, osteoblasts, and proliferating mesenchymal cells from the neural crest.

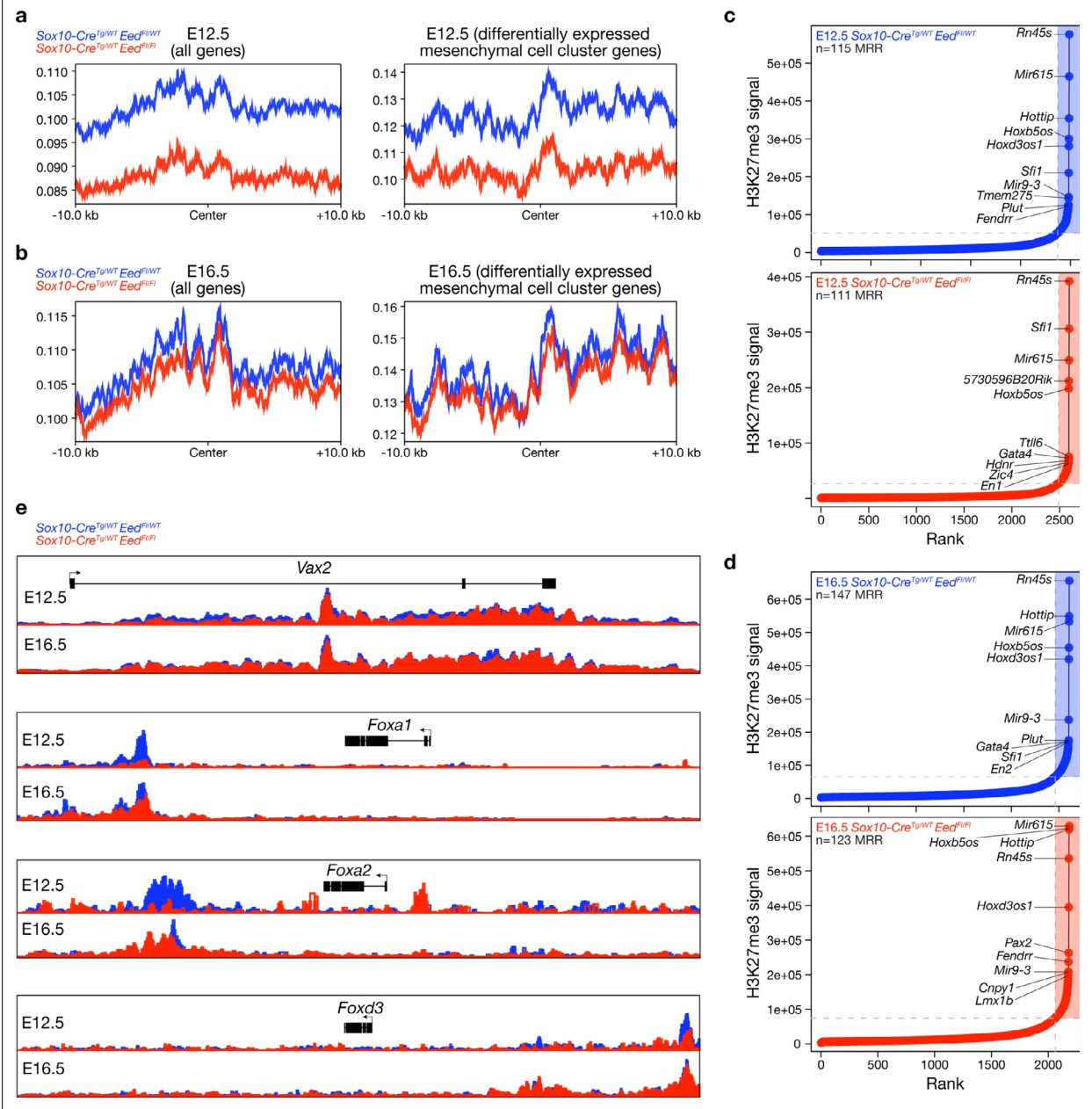

**Figure 5.** *Eed* regulates *H3K27me3* deposition at transcription factor loci that are required for craniofacial development. (**a**) Metaplots of H3K27me3 occupancy within 10 kb of all genes or within 10 kb of differentially expressed mesenchymal cell cluster (C0, C4, C5, C7) from CUT&Tag sequencing of embryonic day (E)12.5 *Sox10-Cre; Eed*^Fl/WT^ (n=3) or *Sox10-Cre; Eed*^Fl/Fl^ (n=3) craniofacial tissues. (**b**) Metaplots of H3K27me3 occupancy within 10 kb of all genes or within 10 kb of differentially expressed mesenchymal cell cluster genes (C0, C4, C5, C7) from CUT&Tag sequencing of E16.5 *Sox10-Cre; Eed*^Fl/WT^ (n=3) or *Sox10-Cre; Eed*^Fl/Fl^ (n=3) craniofacial tissues. (**c**) Distribution of H3K27me3 signal identifying 115 methylation-rich regions (MRRs, shaded) in E12.5 *Sox10-Cre; Eed*^Fl/WT^ (n=3) and 111 MRRs in *Sox10-Cre; Eed*^Fl/Fl^ (n=3) craniofacial tissues. (**d**) Distribution of H3K27me3 signal identifying 147 MRRs in E16.5 *Sox10-Cre; Eed*^Fl/WT^ (n=3) and 123 MRRs in *Sox10-Cre; Eed*^Fl/Fl^ (n=3) craniofacial tissues. (**e**) H3K27me3 tracks at selected loci overlayed from E12.5 or E16.5 *Sox10-Cre; Eed*^Fl/WT^ (n=3 per timepoint) or *Sox10-Cre; Eed*^Fl/Fl^ (n=3 per timepoint) craniofacial tissues.

The online version of this article includes the following figure supplement(s) for figure 5:

**Figure supplement 1.** Gene ontology analyses of H3K27me3 rich regions from embryonic day (E)12.5 or E16.5 *Sox10-Cre; Eed*^Fl/WT^ or *Sox10-Cre; Eed*^Fl/Fl^ craniofacial tissues.

## Discussion

Epigenetic regulation of the neural crest, which differentiates into diverse mesenchymal derivatives, remains incompletely understood. Here, we identify the PRC2 core subunit *Eed* as a potent regulator of craniofacial development after induction of the neural crest. Conditional deletion of *Eed* using *Sox10-Cre* causes severe craniofacial malformations that are consistent with impaired differentiation and proliferation of cells arising from the pharyngeal arches (*Frisdal and Trainor, 2014*). In support of this hypothesis, we observed severe defects in the development of craniofacial mesenchyme-derived tissues (*Figures 1 and 2*) that were consistent with molecular and cellular findings from *Sox10-Cre; Eed^{Fl/Fl}* embryos (*Figures 3–5*). Expression of key regulators of craniofacial osteogenesis, such as *Runx2, Irx3, Irx5, Mecom, Trps1, Ptch1, Pax3,* and *Pax7,* was reduced in *Sox10-Cre; Eed^{Fl/Fl}* heads and primary craniofacial cell cultures compared to controls, and single-cell RNA sequencing showed reduced craniofacial osteoblast differentiation and reduced proliferation of mesenchymal cells in *Sox10-Cre; Eed^{Fl/Fl}* heads compared to controls. H3K27me3 signatures of key developmental transcription factors, including *Barx1, Tfap2a, Vax2, Foxa1, Foxa2, Foxd3,* and others, were markedly reduced in *Sox10-Cre; Eed^{Fl/Fl}* craniofacial tissues compared to controls, suggesting that PRC2 disruption causes widespread mis-regulation of cell fate decision networks in post-migratory neural crest cells. We also identified subtle cardiac phenotypes and a small increase in apoptosis in *Sox10-Cre; Eed^{Fl/Fl}* embryos. In sum, these data suggest that PRC2 may contribute to the development of diverse neural crest-derived tissues but is particularly important for craniofacial mesenchymal cell differentiation and proliferation.

By using *Sox10-Cre*, which marks the migratory neural crest (*Matsuoka et al., 2005*), we bypass potential roles of *Eed* in neural crest cell induction to examine its function in neural crest cell development. The requirement of PRC2 for craniofacial development has been studied in the context of *Ezh2* loss using *Wnt1-Cre* (*Ferguson et al., 2018*; *Schwarz et al., 2014*), but to our knowledge, our study is the first report directly linking *Eed* to differentiation of neural crest-derived mesenchymal cells and the first report of single-cell transcriptomic and epigenomic data underlying these phenotypes.

To date, 15 human missense mutations in *EED* have been reported as pathogenic for Cohen-Gibson syndrome (*Cohen and Gibson, 2016*; *Cohen et al., 2015*; *Goel et al., 2024*). Like Weaver syndrome, which arises from missense mutations in *EZH2* (*Gibson et al., 2012*; *Tatton-Brown et al., 2013*), and Imagawa-Matsumoto syndrome, which arises from missense mutations in *SUZ12* (*Imagawa et al., 2018*), Cohen-Gibson syndrome is a rare congenital syndrome that is associated with craniofacial malformations, advanced bone age, intellectual disability, and developmental delay (*Spellicy et al., 2019*). *EED* mutations in individuals with Cohen-Gibson syndrome cluster in WD40 domains 3 through 5 and are predicted to (1) abolish interaction with EZH2, (2) prevent histone methyltransferase activity, and (3) inhibit H3K27me3 peptide binding. In support of these hypotheses, functional investigations have shown that single nucleotide variants in the WD40 domain of *EED* abolish binding to *EZH2* in vitro (*Denisenko et al., 1998*). Mouse models encoding pathogenic *EZH2* missense variants, which are predicted to result in loss of function of the PRC2 complex, phenocopy Weaver syndrome and cause excess osteogenesis and skeletal overgrowth (*Gao et al., 2023*). In this study, we show that loss of *Eed* after neural crest induction causes the opposite phenotype and results in craniofacial hypoplasia due to impairments in osteogenesis, mesenchymal cell proliferation, and mesenchymal stem cell differentiation. The discrepancy of missense variants in *EED* causing gain of craniofacial osteogenic function in humans versus loss of *Eed* causing loss of craniofacial osteogenic function in mice warrants further study, including investigation using alternative Cre drivers to target different developmental stages, the importance of which is exemplified by limitations associated with some widely used Cre transgenes that target the neural crest (*Lewis et al., 2013*).

Despite these discrepancies, it is notable that many of the phenotypes we observed after loss of *Eed* in the neural crest of mice were similar to previously reported phenotypes after loss of *Ezh2* in the neural crest cell of mice, with the exception of proliferation deficits after loss of *Eed* but not after loss of *Ezh2* (*Ferguson et al., 2018*; *Schwarz et al., 2014*). The temporal and spatial distribution of *Eed* during embryogenesis is well studied, and *Eed* is ubiquitously expressed starting at E5.5, peaks at E9.5, and is downregulated but maintained at a high basal level through E18.5 (*Schumacher et al., 1996*). Although comprehensive analysis of *Eed* expression in neural crest tissues has not been reported (to our knowledge), *Eed* physically and functionally interacts with *Ezh2* (*Sewalt et al., 1998*), which is enriched at a diversity of timepoints throughout all developing craniofacial tissues (*Ferguson*

*et al., 2018*; *Schwarz et al., 2014*). We confirmed enrichment of *Eed* expression in craniofacial tissues throughout development using qPCR, and our single-cell RNA sequencing data failed to reveal any differences in expression of other PRC2 complex members across genotypes. Thus, the available data suggest that there are similarities and that there may be subtle differences in the activity of PRC2 core subunits in developing craniofacial tissues.

## Materials and methods

### Mice

Mice were maintained in the University of California San Francisco (UCSF) pathogen-free animal facility in accordance with the guidelines established by the Institutional Animal Care and Use Committee (IACUC) and Laboratory Animal Resource Center (LARC) protocol AN191840. Mice were maintained in a 70°F, 50% humidity temperature-controlled barrier facilities under a 12–12 hr light cycle with access to food and water ad libitum. Sox10-Cre and *Eed$^{Fl}$* mice were obtained from the Jackson Laboratory (B6;CBA-Tg(Sox10-cre)1Wdr/J, 025807 and B6;129S1-Eed$^{tm1Sho}$/J, 022727, respectively). The presence of the floxed *Eed* allele was determined through standard PCR genotyping using the following primers: 5' GGGACGTGCTGACATTTTCT 3' (forward) and 5' CTTGGGTGGTTTGGCTAAGA 3' (reverse). To generate embryos at specific time points, *Sox10-Cretg$^{tg+}$ Eed$^{Fl/WT}$* mice were bred overnight with *Eed$^{Fl/Fl}$* mice. Females were checked for copulation plugs in the morning, and the presence of a vaginal plug was designated as E0.5.

### Mouse fetal echocardiography

Ultrasound studies were acquired with a Fujifilm Vevo 2100 Imaging system, and instrument specifically designed for lab animal imaging studies. Pregnant females were anesthetized using an isoflurane/oxygen mixture with an isoflurane concentration of 3% during imaging. The pregnancy date (at least p17) was checked to observe the correct development of fetal hearts. Isoflurane was increased until 5% and the abdomen of pregnant females was open surgically to expose the gravid uterus. The number of fetuses was counted, and craniofacial architecture was evaluated for phenotypic confirmation. Once each fetus was defined as mutant or wild type, echocardiography was performed to evaluate left ventricle anatomy and physiology, including left ventricle size and contractility. Three different measurements were obtained in B and M modes, and functional parameters, including heart rate, fraction shortening, ejection fraction, and left ventricle mass, were obtained.

### Whole-mount embryo staining

Embryos were stained as previously described (*Sandell et al., 2018*). In brief, embryos were harvested, and heads were fixed overnight in 4% PFA. Embryos were dehydrated through a methanol series up to 100% before being treated with Dent's bleach (4:1:1, MeOH, DMSO, 30% H$_2$O$_2$) for 2 hr at 25°C. Embryos were serially rehydrated in methanol up to 25%, washed with PBST, and stained with DAPI solution (1:40,000, Thermo Scientific, cat# 62248) overnight at 4°C. Samples were mounted in low melt agarose for imaging.

### Whole-mount skeletal staining

Embryos were stained as previously described (*Rigueur and Lyons, 2013*). In brief, embryos were dissected in cold PBS and scalded in hot water to facilitate the removal of eyes, skin, and internal organs. Embryos were fixed in 95% ethanol overnight, then subsequently incubated in acetone overnight. Embryos were stained with Alcian blue solution (Newcomer Supply, cat# 1300B) for 8 hr, washed in 70% ethanol, and destained in 95% ethanol. Embryos were cleared in 95% ethanol and 1% KOH and stained in Alizarin Red S solution (SCBT, cat# 130-22-3) for 4 hr. Embryos were stored in glycerol for imaging.

### Micro-computed tomography

microCT studies were acquired with a Perkin Elmer's Quantum GX2 microCT Imaging system, an instrument specifically designed for lab animal imaging studies. The samples were imaged in Eppendorf tubes containing preservative media, which were placed on the scanning bed and fixed with tape to minimize movement. Acquisition parameters were set as follows: FOV 10 mm, acquisition time

14 min, current voltage 70 kV, amperage 114 µA. Each study was composed of a 512×512×512 voxels matrix with a spatial resolution of 0.018 mm$^3$. Study reconstruction was based on Feldkamp's method using instrument dedicated software. For image analysis, tridimensional renders were obtained using 3D SLICER software. The threshold range applied for bone segmentation was 300–1400 Hounsfield units. For the segmentation and visualization of individual frontal bones, cranial base, and mandible, Avizo Lite software (v9.1.1) was used. The threshold range applied was 570–1570 Hounsfield units.

## Cryosectioning and H&E staining

Embryos were fixed in 4% PFA overnight at 4°C and subjected to a sucrose gradient from 15% to 30% before embedding in OCT (Tissue-Tek, cat# 4583) and storage at –80°C. Embryos were sectioned at 25 µm on a Leica CM1850 cryostat. Slides were immediately subjected to H&E staining as previously described (*Cardiff et al., 2014*) or stored at –80°C for immunofluorescence.

## Immunofluorescence

Immunofluorescence of cryosections was performed on glass slides. Immunofluorescence for primary cells was performed on glass coverslips. Primary cells were fixed in 4% PFA for 10 min at room temperature. Antibody incubations were performed in blocking solution (0.1% Triton X-100, 1% BSA, 10% donkey serum) for 2 hr at room temperature or overnight at 4°C. Slides or coverslips were mounted in ProLong Diamond Antifade Mountant (Thermo Scientific, cat# P36965). Primary antibodies used in this study were anti-Sox9 (1:500, Proteintech, cat# 67439-1-Ig), anti-Sox9 (1:500, Abcam, cat# ab185230), anti-Runx2 (1:500, Cell Signaling, cat# D1L7F), anti-Ki67 (1:2000, Abcam, cat# ab15580), anti-Vimentin (1:1000, Abcam, cat# ab8069), anti-BrdU (1:1000, Abcam, cat# ab6326), anti-Sox10 (1:500, Cell Signaling, cat# D5V9L), anti-ALPL (1:1000, Invitrogen, cat# PA5-47419), anti-Eed (1:1000, Cell Signaling, cat# E4L6E), and anti-Cleaved Caspase-3 (Asp175) (1:500, Cell Signaling, cat#9661). Secondary antibodies used in this study were Donkey anti-Rabbit IgG (H+L) Alexa Fluor Plus 555 (1:1000, Thermo Scientific, cat# A32794), Donkey anti-Rabbit IgG (H+L) Alexa Fluor 488 (1:1000, Thermo Scientific, cat# A21206), and Donkey anti-Rabbit IgG (H+L) Alexa Fluor Plus 568 (1:1000, Thermo Scientific, cat# A10042). DAPI solution (1:5000, Thermo Scientific, cat# 62248) was added to secondary antibody solutions. Slides or coverslips were mounted in ProLong Diamond Antifade Mountant (Thermo Scientific, cat# P36965) and cured overnight before imaging.

## BrdU staining

Timed mating dams were subjected to intraperitoneal injection of 100 mg/kg sterile BrdU in PBS (Abcam, cat# ab142567). After 4 hr, mice were euthanized, and embryos were harvested in cold PBS. Embryos were fixed in 4% PFA overnight at 4°C and cryosectioned. Sections were incubated with 1 M HCl for 2 hr then 0.1 M sodium borate buffer for 15 min before being subjected to standard immunofluorescence.

## Microscopy and image analysis

Whole-mount skeletal and H&E stains were imaged using a Zeiss Stemi 305 Stereo Zoom microscope running Zeiss Blue v2.0. Whole-mount embryo DAPI stains were imaged using an upright Zeiss Axio Imager Z2 running AxioVision v4.0. Immunofluorescence images were obtained using a Zeiss LSM800 confocal laser scanning microscope running Zen Blue v2.0. Quantification of embryo measurements and immunofluorescence staining intensities was performed in ImageJ using standard thresholding and measurement of signal integrated density. Signal to nuclei normalization was performed by dividing signal intensity by DAPI signal per image. Quantification was performed using n>6 images per embryo. Proliferation index was calculated by dividing the number of Ki67+ or BrdU+ nuclei by the total number of DAPI-stained nuclei per image.

## Primary craniofacial cell culture

Timed mating embryos were dissected on ice-cold PBS and heads were removed. Brains and eyes were removed from each head, and the remaining craniofacial structures were minced with a sterile blade on a glass surface. Tissue was enzymatically dissociated in 3 mg/ml Collagenase Type 7 (Worthington, cat# CLS-6) in HBSS for 2–3 hr at 37°C with frequent trituration using an Eppendorf ThermoMixer. Dissociated cells were filtered through a 70 µM MACS smart strainer (Miltenyi Biotec, cat#

130-110-916) and either plated on coverslips or cell culture dishes. Primary craniofacial cells were grown in DMEM with 10% FBS. For RNA extraction and immunofluorescence, cells were harvested 1 day following initial dissociation.

## Single-cell RNA sequencing

Matched littermate timed embryos were dissected on ice-cold PBS. Embryo heads were removed and minced using sterile razor blades on a glass surface. Tissue was enzymatically dissociated in 3 mg/ml Collagenase Type 7 (Worthington, cat# CLS-6) in HBSS for 2–3 hr at 37°C with frequent trituration using an Eppendorf ThermoMixer. The quality of the dissociation was frequently monitored by looking at the cell suspension on an automated cell counter. Cells were spun down at 350 rcf, resuspended in MACS BSA stock solution (Miltenyi Biotec, cat# 130-091-376), and serially filtered through 70 μm and 40 μm MACS smart strainers (Miltenyi Biotec, cat# 130-110-916). Dissociation quality checks, cell viability, and counting were performed using an Invitrogen Countess 3 automated cell counter. 10,000 cells were loaded per single-cell RNA sequencing sample. Single-cell RNA sequencing libraries were generated using the Chromium Single Cell 3' Library & Gel Bead Kit v3.1 on a 10x Genomics Chromium controller using the manufacturer's recommended default protocol and settings.

## Single-cell RNA sequencing analysis

Library demultiplexing, read alignment to the mouse reference genome mm10, and unique molecular identifier (UMI) quantification was performed in Cell Ranger v7.2.0. Cells with greater than 200 unique genes were retained for analysis. Data were normalized and variance-stabilized by SCTransform in Seurat v5.0 (*Hao et al., 2024*). UMAP and cluster analysis were performed using the Seurat function RunUMAP with parameters of mindist = 0.7, res = 0.4, dims = 1:30. Cluster markers were identified using the Seurat function FindAllMarkers with parameters min.pct=0.25, thresh.use=0.25. Differential gene expression analysis was performed using DElegate v1.1.0 (*Hafemeister and Halbritter, 2023*) using the DESeq2 method, with biological replicate animals (n=3 per genotype) as the replicate_ column parameter. FeaturePlots were generated using SCPubr v1.1.2 (*Blanco-Carmona, 2022*), and scale bars represent log2 UMIs corrected by SCTransform.

## CUT&Tag sequencing

E12.5 or E16.5 timed mating embryos were dissected on ice-cold PBS. Craniofacial tissues were micro-dissected under a Leica Stereoscope, minced, and weighed such that 30 mg of craniofacial tissue was harvested per embryo. Tissue was then processed, and libraries were generated according to the manufacturer's instructions using an end-to-end CUT&Tag-IT Assay Kit (Active Motif, cat# 53170). In brief, nuclei were mechanically isolated in a Dounce homogenizer with lysis buffer containing protease inhibitors and filtered through a 40 μm strainer. Nuclei were pelleted, washed, and counted such that each reaction was normalized to 500,000 nuclei per embryo. To enable quantitative normalization across samples, 20,000 *Drosophila melanogaster* spike-in nuclei (Active Motif, cat# 53173) were added prior to bead conjugation. Nuclei were then bound to pre-equilibrated concanavalin A-coated magnetic beads and incubated with a primary antibody against H3K27me3 (Active Motif, cat# 39055) along with a *Drosophila*-specific anti-H2Av spike-in antibody (Active Motif, cat# 61686) overnight at 4°C in antibody binding buffer containing 0.05% digitonin, 2 mM EDTA, and 0.1% BSA. Following washes with digitonin-containing buffer, samples were incubated with species-appropriate secondary antibodies for 1 hr at room temperature and subsequently exposed to a pre-loaded pA-Tn5 trans-posome complex (1:20 dilution) in Dig-300 Buffer for 1 hr to enable tethering. Tagmentation was initiated by the addition of 10 mM $MgCl_2$ at 37 °C for 1 hr, after which the reaction was terminated with EDTA, 0.1% SDS, and proteinase K digestion at 55°C for 1 hr. DNA was purified using magnetic bead-based cleanup and amplified with i7 and i5 indexed primers using standard PCR amplification. Libraries were cleaned with SPRI beads and quality-assessed on an Agilent Tape Station. Sequencing was performed on an Illumina NovaSeq X with paired-end reads targeting 25 million reads per sample.

## CUT&Tag sequencing analysis

Raw H3K27me3 reads were processed using the nf-core Cut&Run pipeline (v1.0.0) and aligned to the mm10 genome. Signal tracks were generated as bigWig files. Replicate bigWigs were merged, averaged, and CPM-normalized using deepTools bamCompare (v3.5.0) with options --normalizeUsing

CPM, −−scaleFactorsMethod None, and −−binSize 50. Peaks were called on merged bigWigs using SEACR (v1.3) in stringent mode with an FDR threshold of 0.001. Signal metaplots were generated using deepTools computeMatrix and plotProfile with −−referencePoint center and flanking windows of ±10 kb. Metaplots were generated for all protein-coding genes and for selected differentially expressed genes (DEGs). DEG regions were defined by identifying DEGs from differential expression analyses and retrieving their transcription start site (TSS) coordinates using biomaRt (Ensembl release 106). TSS regions were expanded by ±10 kb, and BED files were generated by excluding genes located on non-autosomal chromosomes (MT, X, Y). The enhancer ranking and super-enhancer identification strategy was adapted from previously described methods (*Lovén et al., 2013*; *Whyte et al., 2013*). Peaks overlapping TSS were removed using BEDTools intersect (v2.31.1) with the -v option. MRRs were identified using ROSE (*Lin et al., 2020*) with default settings, and putative MRRs ranked by area under the curve (AUC) from bigWig signal. Inflection points in AUC-ranked plots were used to define MRRs. MRRs were annotated to the nearest RefSeq gene using the TxDb.Mmusculus. UCSC.mm-10.knownGene (v3.10.0) and org.Mm.eg.db (v3.18.0) packages in R. Hockey-stick plots were generated by plotting H3K27me3 signal versus rank.

## Statistics

All experiments were performed with independent biological replicates and repeated, and statistics were derived from biological replicates. Biological replicates are indicated in each figure panel or figure legend. No statistical methods were used to predetermine sample sizes, but sample sizes in this study are similar or larger to those reported in previous publications. Data distribution was assumed to be normal, but this was not formally tested. Investigators were blinded to conditions during data collection and analysis. Bioinformatic analyses were performed blind to clinical features, outcomes, and molecular characteristics. The samples used in this study were nonrandomized with no intervention, and all samples were interrogated equally. Thus, controlling for covariates among samples was not relevant. No data points were excluded from the analyses.

## Code availability

No custom software, tools, or packages were used. The open-source software, tools, and packages used for data analysis in this study are referenced in the methods where applicable and include Cell Ranger v7.2.0, Seurat v5.0, DElegate v1.1.0, SCPubr v1.1.2, nf-core Cut&Run (v1.0.0), deepTools bamCompare (v3.5.0), SEACR (v1.3), BEDTools intersect (v2.31.1), TxDb.Mmusculus.UCSC.mm-10. knownGene (v3.10.0), and org.Mm.eg.db (v3.18.0).

## Additional information

### Funding

| Funder | Grant reference number | Author |
|---|---|---|
| National Institutes of Health | T32 CA151022 | Tim Casey-Clyde |
| National Institutes of Health | R35 DE031926 | Jeffrey Ohmann Bush |
| National Institutes of Health | R21 HD106238 | David R Raleigh |
| Department of Defence | NFRP NF200021 | David R Raleigh |

The funders had no role in study design, data collection and interpretation, or the decision to submit the work for publication.

### Author contributions

Tim Casey-Clyde, Conceptualization, Data curation, Formal analysis, Investigation, Visualization, Methodology, Writing – original draft, Writing – review and editing; S John Liu, Formal analysis, Investigation, Visualization, Writing – review and editing; Angelo Pelonero, Formal analysis, Investigation; Juan Antonio Camara Serrano, Investigation, Visualization; Camilla Teng, Investigation, Methodology;

Yoon-Gu Jang, Investigation; Harish N Vasudevan, Conceptualization, Project administration; Arun Padmanabhan, Formal analysis, Supervision, Project administration, Writing – review and editing; Jeffrey Ohmann Bush, Supervision, Writing – original draft, Project administration; David R Raleigh, Formal analysis, Supervision, Funding acquisition, Investigation, Visualization, Writing – original draft, Project administration, Writing – review and editing

### Author ORCIDs
Jeffrey Ohmann Bush ⬥ https://orcid.org/0000-0002-6053-8756
David R Raleigh ⬥ https://orcid.org/0000-0001-9299-8864

### Ethics
Mice were maintained in the University of California San Francisco (UCSF) pathogen-free animal facility in accordance with the guidelines established by the Institutional Animal Care and Use Committee (IACUC) and Laboratory Animal Resource Center (LARC) protocol AN191840.

Reviewer #2 (Public review): https://doi.org/10.7554/eLife.100159.3.sa1
Author response https://doi.org/10.7554/eLife.100159.3.sa2

## Additional files

### Supplementary files
Supplementary file 1. Cell count per embryo and per cell cluster from single-cell RNA sequencing.

Supplementary file 2. Single-cell RNA sequencing cluster marker genes.

Supplementary file 3. Differential gene expression between C4 and C0 from single-cell RNA sequencing.

Supplementary file 4. Differential gene expression between C5 and C0 from single-cell RNA sequencing.

Supplementary file 5. Differential gene expression between C4 and C5 from single-cell RNA sequencing.

Supplementary file 6. Differential gene expression between genotypes in C0, C4, C5, and C7 from single-cell RNA sequencing.

Supplementary file 7. Methylation rank and nearest gene coordinates from CUT&Tag H3K27me3 sequencing of E12.5 Sox10-Cre; Eed$^{Fl/WT}$ craniofacial tissues.

Supplementary file 8. Methylation rank and nearest gene coordinates from CUT&Tag H3K27me3 sequencing of E12.5 Sox10-Cre; Eed$^{Fl/Fl}$ craniofacial tissues.

Supplementary file 9. Methylation rank and nearest gene coordinates from CUT&Tag H3K27me3 sequencing of E16.5 Sox10-Cre; Eed$^{Fl/WT}$ craniofacial tissues.

Supplementary file 10. Methylation rank and nearest gene coordinates from CUT&Tag H3K27me3 sequencing of E16.5 Sox10-Cre; Eed$^{Fl/Fl}$ craniofacial tissues.

MDAR checklist

### Data availability
Single-cell RNA sequencing data and CUT&Tag sequencing data that are reported in this manuscript have been deposited in the NCBI Sequence Read Archive under PRJNA1077750 (https://www.ncbi.nlm.nih.gov/bioproject/PRJNA1077750) and GSE302601 (https://www.ncbi.nlm.nih.gov/geo/query/acc.cgi?&acc=GSE302601).

The following datasets were generated:

| Author(s) | Year | Dataset title | Dataset URL | Database and Identifier |
|---|---|---|---|---|
| Tim C, Arun P, Angelo P, David R, John LS, Juan C, Camilla T, Yoon-Gu J, Harish V, Jeffrey B | 2025 | Eed controls craniofacial osteoblast differentiation and mesenchymal proliferation from the neural crest | https://www.ncbi.nlm.nih.gov/geo/query/acc.cgi?&acc=GSE302601 | NCBI Gene Expression Omnibus, GSE302601 |
| University of California, San Francisco | 2024 | Eed controls mesenchymal stem cell differentiation in craniofacial derivatives of the neural crest | https://www.ncbi.nlm.nih.gov/bioproject/PRJNA1077750 | NCBI BioProject, PRJNA1077750 |

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
