## [Editor Report · eLife Assessment]

In this **valuable** study, the authors used an elegant genetic approach to delete EED at the post-neural crest induction stage. The usage of the single-cell RNA-seq analysis method is extremely suitable to determine changes in the cell type-specific gene expression during development. Results backed by **solid** evidence demonstrate that Eed is required for craniofacial osteoblast differentiation and mesenchymal proliferation after the induction of the neural crest.

---

## [Referee Report · Reviewer #2 (Public review)]

Summary:

The role of PRC2 in post neural crest induction was not well understood. This work developed an elegant mouse genetic system to conditionally deplete EED upon SOX10 activation. Substantial developmental defects were identified for craniofacial and bone development. The authors also performed extensive single-cell RNA sequencing to analyze differentiation gene expression changes upon conditional EED disruption.

Strengths:

(1) Elegant genetic system to ablate EED post neural crest induction.

(2) Single-cell RNA-seq analysis is extremely suitable for studying the cell type specific gene expression changes in developmental systems.

Original Weaknesses:

(1) Although this study is well designed and contains state-of-art single cell RNA-seq analysis, it lacks the mechanistic depth in the EED/PRC2-mediated epigenetic repression. This is largely because no epigenomic data was shown.

(2) The mouse model of conditional loss of EZH2 in neural crest has been previously reported, as the authors pointed out in the discussion. What is novelty in this study to disrupt EED? Perhaps a more detailed comparison of the two mouse models would be beneficial.

(3) The presentation of the single-cell RNA-seq data may need improvement. The complexity of the many cell types blurs the importance of which cell types are affected the most by EED disruption.

(4) While it's easy to identify PRC2/EED target genes using published epigenomic data, it would be nice to tease out the direct versus indirect effects in the gene expression changes (e.g Fig. 4e)

Comments on latest version:

The authors have addressed weaknesses 2 and 3 of my previous comment very well. For weaknesses 1 and 4, the authors have added a main Fig 5 and its associated supplemental materials, which definitely strengthen the mechanistic depth of the story. However, I think the audience would appreciate if the following questions/points could be further addressed regarding the Cut&Tag data (mostly related to main Figure 5):

(1) The authors described that Sox10-Cre would be expressed at E8.75, and in theory, EED-FL would be ablated soon after that. Why would E16.5 exhibit a much smaller loss in H3K27me3 compared to E12.5? Shouldn't a prolong loss of EED lead to even worse consequence?

(2) The gene expression change at E12.5 upon loss of EED (shown in Fig. 4h) seems to be massive, including many PRC2-target genes. However, the H3K27me3 alteration seems to be mild even at E12.5. Does this infer a PRC2 or H3K27 methylation - independent role of EED? To address this, I suggest the authors re-consider addressing my previously commented weakness #4 regarding the RNA-seq versus Cut&Tag change correlation. For example, a gene scatter plot with X-axis of RNA-seq changes versus Y-axis of H3K27me3 level changes.

(3) The CUT&Tag experiments seem to contain replicates according to the figure legend, but no statistical analysis was presented including the new supplemental tables. Also, for Fig. 5c-d, instead of showing the MRR in individual conditions, I think the audience would really want to know the differential MRR between Fl/WT and Fl/Fl. In other words, how many genes/ MRR have statistically lower H3K27me3 level upon EED loss.

---

## [Author Response]

The following is the authors’ response to the original reviews.

**Reviewer #1 (Public review):**
Epigenetic regulation complex (PRC2) is essential for neural crest specification, and its misregulation has been shown to cause severe craniofacial defects. This study shows that Eed, a core PRC2 component, is critical for craniofacial osteoblast differentiation and mesenchymal proliferation after neural crest induction. Using mouse genetics and single-cell RNA sequencing, the researcher found that conditional knockout of Eed leads to significant craniofacial hypoplasia, impaired osteogenesis, and reduced proliferation of mesenchymal cells in post-migratory neural crest populations.Overall, the study is superficial and descriptive. No in-depth mechanism was analyzed and the phenotype analysis is not comprehensive.

We thank the reviewer for sharing their expertise and for taking the time to provide helpful suggestions to improve our study. We are gratified that the striking phenotypes we report from Eed loss in post-migratory neural crest craniofacial tissues were appreciated. The breadth and depth of our phenotyping techniques, including skeletal staining, micro-CT, echocardiogram, immunofluorescence, histology, and primary craniofacial cell culture provide comprehensive data in support our hypothesis that PRC2 is required for epigenetic control of craniofacial osteoblast differentiation. To provide mechanistic data in support of this hypothesis, we have now performed CUT&Tag H3K27me3 chromatin profiling on nuclei harvested from E12.5 or E16.5 Sox10-Cre Eed^Fl/WT^ and Sox10-Cre Eed^Fl/Fl^ craniofacial tissue. These new data, which are presented in Fig. 5, Supplementary Fig. 9, and Supplementary Tables 7-10 of our revised manuscript, validate our hypothesis that epigenetic regulation of chromatin architecture downstream of PRC2 activity underlies craniofacial osteoblast differentiation. In particular, we now show that Eed-dependent H3K27me3 methylation is associated with correct temporal expression of transcription factors that are necessary for craniofacial differentiation and patterning, such as including Msx1, Pitx1, Pax7, which were initially nominated by single-cell RNA sequencing of E12.5 Sox10-Cre Eed^Fl/WT^ and Sox10-Cre Eed^Fl/Fl^ craniofacial tissues in Fig. 4, Supplementary Fig. 5-7, and Supplementary Tables 1-6.

**Reviewer #2 (Public review):**
Summary:The role of PRC2 in post-neural crest induction was not well understood. This work developed an elegant mouse genetic system to conditionally deplete EED upon SOX10 activation. Substantial developmental defects were identified for craniofacial and bone development. The authors also performed extensive single-cell RNA sequencing to analyze differentiation gene expression changes upon conditional EED disruption.Strengths:(1) Elegant genetic system to ablate EED post neural crest induction.(2) Single-cell RNA-seq analysis is extremely suitable for studying the cell type-specific gene expression changes in developmental systems.

We thank the reviewer for their generous and helpful comments on our study. We are happy that our mouse genetic and single-cell RNA sequencing approaches were appropriate in pairing the craniofacial phenotypes we report with distinct gene expression changes in post-migratory neural crest tissues upon Eed deletion.

Weaknesses:(1) Although this study is well designed and contains state-of-the-art single-cell RNA-seq analysis, it lacks the mechanistic depth in the EED/PRC2-mediated epigenetic repression. This is largely because no epigenomic data was shown.

Thank you for this suggestion. As described in response to Reviewer #1, we have now performed CUT&Tag H3K27me3 chromatin profiling on nuclei harvested from E12.5 or E16.5 Sox10-Cre Eed^Fl/WT^ and Sox10-Cre Eed^Fl/Fl^ craniofacial tissues to provide mechanistic epigenomic data in support of our hypothesis that hat PRC2 is required for craniofacial osteoblast differentiation. These new data, which are presented in Fig. 5, Supplementary Fig. 9, and Supplementary Tables 7-10 of our revised manuscript, integrate genome-wide and targeted metaplot visualizations across genotypes with in-depth analyses of methylation rich regions and genes associated with methylation rich loci. Broadly, these new data reveal that changes in H3K27me3 occupancy correlate with gene expression changes from single-cell RNA sequencing of E12.5 Sox10-Cre Eed^Fl/WT^ and Sox10-Cre Eed^Fl/Fl^ craniofacial tissues in Fig. 4, Supplementary Fig. 5-7, and Supplementary Tables 1-6.

(2) The mouse model of conditional loss of EZH2 in neural crest has been previously reported, as the authors pointed out in the discussion. What is novel in this study to disrupt EED? Perhaps a more detailed comparison of the two mouse models would be beneficial.

We acknowledge and cite the study the reviewer has indicated (Schwarz et al. Development 2014) in our initial and revised manuscripts. This elegant investigation uses Wnt1-Cre to delete Ezh2 and reports a phenotype similar to the one we observed with Sox10-Cre deletion of Eed, but our study adds depth to the understanding of PRC2’s vital role in neural crest development by ablating Eed, which has a unique function in the PRC2 complex by binding to H3K27me3 and allosterically activating Ezh2. In this sense, our study sheds light on whether phenotypes arising from deletion of Eed, the PRC2 “reader”, differ from phenotypes arising from deletion of Ezh2, the PRC2 “writer”, in neural crest derived tissues. Moreover, we provide the first single-cell RNA sequencing and epigenomic investigations of craniofacial phenotypes arising from PRC2 activity in the developing neural crest. Due to limitations associated with the Wnt1-Cre transgene (Lewis et al. Developmental Biology 2013), which targets pre-migratory neural crest cells, our investigations used Sox10Cre, which targets the migratory neural crest and is completely recombined by E10.5. We have included a detailed comparison of these mouse models in the Discussion section of our revised manuscript, and we thank the reviewer for this thoughtful suggestion.

(3) The presentation of the single-cell RNA-seq data may need improvement. The complexity of the many cell types blurs the importance of which cell types are affected the most by EED disruption.

We thank the reviewer for the opportunity to improve the presentation of our single-cell RNA sequencing data. In response, we have added Supplementary Fig. 8 to our revised manuscript, which shows the cell clusters most affected by EED disruption in UMAP space across genotypes. Because we wanted to capture the fill diversity of cell types underlying the phenotypes we report, we did not sort Sox10+ cells (via FACS, for example) from craniofacial tissues before single-cell RNA sequencing. Our resulting single-cell RNA sequencing data are therefore inclusive of a diversity of cell types in UMAP space, and the prevalence of many of these cell types was unaffected by epigenetic disruption of neural crest derived tissues. The prevalence of the cell clusters that are most affected across genotypes and which are most relevant to our analyses of the developing neural crest are shown in Fig. 4c (and now also in Supplementary Fig. 8), including C0 (differentiating osteoblasts), C4 (mesenchymal stem cells), C5 (mesenchymal stem cells), and C7 (proliferating mesenchymal stem cells). Marker genes and pseudobulked differential expression analyses across these clusters are shown in Fig. 4d and Fig. 4e-h, respectively.

(4) While it's easy to identify PRC2/EED target genes using published epigenomic data, it would be nice to tease out the direct versus indirect effects in the gene expression changes (e.g Figure 4e).

We agree with the reviewer that the single-cell RNA sequencing data in our initial submission do not provide insight into direct versus indirect changes in gene expression downstream of PRC2. In contrast, the CUT&Tag chromatin profiling data that we have generated for this revision provides mechanistic insight into H3K27me3 occupancy and direct effects on gene expression resulting from PRC2 inactivation in our mouse models.

**Reviewing editor comments:**
The following are recommended as essential revisions(1) The study is overall superficial and primarily descriptive, lacking in-depth mechanistic analysis and comprehensive phenotype evaluation.

Please see responses to Reviewer #1 and Reviewer #2 (weaknesses 1 and 4) above.

(2) The authors did not investigate the temporal and spatial expression of Eed during cranial neural crest development, which is crucial for explaining the observed phenotypes.

The temporal and spatial expression of Eed during embryogenesis is well studied. Eed is ubiquitously expressed starting at E5.5, peaks at E9.5, and is downregulated but maintained at a high basal expression level through E18.5 (Schumacher et al. Nature 1996). Although comprehensive analysis of Eed expression in neural crest tissues has not been reported (to our knowledge), Eed physically and functionally interacts with Ezh2 (Sewalt et al. Mol Cell Biol 1998), which is enriched at a diversity of timepoints throughout all developing craniofacial tissues (Schwarz et al. Development 2014). In our study, we confirmed enrichment of Eed expression in craniofacial tissues throughout development using QPCR, and have provided a more detailed description of these published and new findings in the Discussion section of our revised manuscript.

(3) There is no apoptosis analysis provided for any of the samples.

We evaluated the presence of apoptotic cells in E12.5 craniofacial sections using immunofluorescence for Cleaved Caspase 3 in Supplementary Fig. 3d. Although we found a modest increase in the labeling index of apoptotic cells, there was insufficient evidence to conclude that apoptosis is a substantial factor in craniofacial hypoplasia resulting from Eed loss in post-migratory neural crest craniofacial tissues. We have clarified these findings in the Results and Discussion sections of our revised manuscript.

(4) As Eed is a core component of the PRC2 complex, were any other components altered in the Eed cKO mutant? How does Eed regulation influence osteogenic differentiation and proliferation through known pathways?

We thank the editors for this thoughtful inquiry. Although we did not specifically investigate expression or stability of other PRC2 components in Eed conditional mutants, and little is known about how Eed regulates osteogenic differentiation or proliferation through any pathway, our single-cell RNA sequencing data presented in Fig. 4, Supplementary Fig. 5-7, and Supplementary Tables 1-6 provide a significant conceptual advance with mechanistic implications for understanding bone development downstream of Eed and do not reveal any alterations in the expression of other PRC2 components across genotypes. We have clarified these important details in the Discussion section of our revised manuscript.

(5) The authors may compare the Eed cKO phenotype with that of the previous EZH2 cKO mouse model since both Eed and EZH2 are essential subunits of PRC2.

Please see responses to editorial comment 2 above and the last paragraph of the Discussion section of our revised manuscript for comparisons between Eed and Ezh2 knockout phenotypes.